# An Aggrephagy-Related LncRNA Signature for the Prognosis of Pancreatic Adenocarcinoma

**DOI:** 10.3390/genes14010124

**Published:** 2023-01-02

**Authors:** Xueyuan Huang, Hao Chi, Siqi Gou, Xiyuan Guo, Lin Li, Gaoge Peng, Jinhao Zhang, Jiayu Xu, Siji Nian, Qing Yuan

**Affiliations:** 1Immune Mechanism and Therapy of Major Diseases of Luzhou Key Laboratory, Public Center of Experimental Technology, School of Basic Medical Sciences, Southwest Medical University, Luzhou 646000, China; 2Clinical Medical College, Southwest Medical University, Luzhou 646000, China; 3School of Stomatology, Southwest Medical University, Luzhou 646000, China; 4Statistics Department, School of Science, Minzu University of China, Beijing 100081, China

**Keywords:** aggrephagy, immunotherapy, LncRNA, PAAD, prognostic signature

## Abstract

Pancreatic adenocarcinoma (PAAD) is a common, highly malignant, and aggressive gastrointestinal tumor. The conventional treatment of PAAD shows poor results, and patients have poor prognosis. The synthesis and degradation of proteins are essential for the occurrence and development of tumors. Aggrephagy is a type of autophagy that selectively degrades aggregated proteins. It decreases the formation of aggregates by degrading proteins, thus reducing the harm to cells. By breaking down proteins, it decreases the formation of aggregates; thus, minimizing damage to cells. For evaluating the response to immunotherapy and prognosis in PAAD patients, in this study, we developed a reliable signature based on aggrephagy-related genes (ARGs). We obtained 298 AGGLncRNAs. Based on the results of one-way Cox and LASSO analyses, the lncRNA signature was constructed. In the risk model, the prognosis of patients in the low-risk group was noticeably better than that of the patients in the high-risk group. Additionally, the ROC curves and nomograms validated the capacity of the risk model to predict the prognosis of PAAD. The patients in the low-risk and high-risk groups showed considerable variations in functional enrichment and immunological analysis. Regarding drug sensitivity, the low-risk and high-risk groups had different half-maximal inhibitory concentrations (IC50).

## 1. Introduction

Pancreatic adenocarcinoma (PAAD) is a major pancreatic cancer and one of the top causes of mortality in people with gastrointestinal cancers. Its highly malignant and aggressive nature results in high morbidity and mortality every year around the world. Around 331,000 fatalities are caused each year by pancreatic cancer [1,2]. PAAD patients often develop local infiltration and distant metastasis, and these make conventional treatment strategies (e.g., surgery, chemotherapy, radiotherapy, etc.) useless. Thus, the five-year survival rate of PAAD patients is very poor (only 9% in the USA) [2,3,4,5]. The TNM stages and histological grading form the basic framework for patient treatment planning since they are strongly associated with the prognosis of malignancies [6]. Early signs of PAAD are not known, and most patients are already at an advanced stage when PAAD is diagnosed. Metastasis to distant organs, such as the liver, lungs, and bones, might also occur. The clinicopathological features alone are not effective in guiding the treatment of patients with PAAD to increase their survival. However, since most PAAD patients exhibit different clinicopathological features, this suggests that the traditional pathological grading and staging are not very accurate for guiding the treatment of PAAD patients [7,8]. To improve the quality of life of PAAD patients and provide a foundation for precision medicine therapy, identifying novel prognostic biomarkers is necessary.

Protein aggregates are produced because of incorrect folding or misfolding caused by mutations, inadequate translation, inappropriate post-translational modifications, and oxidative stress [9]. The aggregation of misexpressed proteins often indicates cancer development. Aggrephagy is a kind of selective autophagy that detects and destroys protein aggregation [10]. Aggrephagy regulates viral infection [11], preeclampsia [12], neurodegenerative diseases, heart failure [13], and other diseases. PAAD tissues and cells have higher levels of autophagy-related gene expression and activity [14,15,16]. Autophagy exhibits bidirectional regulation of tumorigenesis. For example, low levels of autophagy promote early cancer development, whereas high levels of autophagy promote the survival of tumor cells in a nutrient-deficient microenvironment [17]. Whether or not aggrephagy, as a specific selective autophagy, plays the same regulatory role as autophagy in PAAD, there are not enough studies to confirm this.

Long intergenic non-coding RNA (lncRNA) is an autonomously transcribed non-coding RNA over 200 nucleotides long [18]. The dysregulation of lncRNA affects functions, such as the proliferation of cells, anti-apoptosis, promotion of metastasis, and evasion of tumor suppressors, resulting in tumorigenesis and progression [19,20]. Recent studies have shown that lncRNA, a regulator of autophagy, can help in regulating various steps related to the formation and maturation of autophagosomes [21,22,23]. The effect of AGGLncRNAs on the assessment and treatment of PAAD is unclear. Therefore, in this study, we investigated the association between the expression patterns of AGGLncRNAs and the prognosis of PAAD patients.

We screened four reliable AGGLncRNAs (CASC8, LINC01091, LINC02600, and PAN3−AS1) from the TCGA-PAAD cohort to develop a new risk prognostic model. We also determined the associations between this risk-scoring model and chemotherapy sensitivity, immunotherapy, and the immune microenvironment. We aimed to generate new clinical guidelines to enhance patient treatment programs and improve the quality of survival for PAAD patients, as well as evaluate the importance of AGGLncRNAs in determining the prognosis of PAAD patients through a thorough examination of genomic data.

## 2. Materials and Methods

### 2.1. Data Source and Processing

For PAAD patients, we obtained information on the transcriptome and the accompanying clinical data from the TCGA database (https://portal.gdc.cancer.gov/ (accessed on 3, October, 2022). The data on patients in the M stage were excluded due to large variations. We used the Strawberry Perl software to distinguish between mRNA and lncRNA, and we used lncRNA-related data in this study. The clinical data included information on the age, gender, tumor stage, and TNM stage of the patients. The aggrephagy-related genes were obtained from a website (https://www.gsea-msigdb.org/gsea/msigdb/cards/REACTOME_AGGREPHAGY (accessed on 3, October, 2022). The information on the ARGs is provided in Appendix A.

### 2.2. Screening of AGGLncRNAs

Using the R package “limma”, we performed the analysis and retrieved the genes that exhibited differential expression. We then developed the AGGLncRNAs by determining the relationship between ARGs and the expression of lncRNAs, with the correlation coefficient set to >0.4 and the *p*-value set to <0.001. The AGGLncRNAs with a |log2 fold change| > 2 and a false discovery rate (FDR) < 0.05 were considered to be substantially differentially expressed. We used the R packages “limma”, “tidyverse”, “ggplot2”, and “ggExtra” to construct the Sankey plots for visualizing the results.

### 2.3. Development and Verification of an AGGLncRNA Prognostic Model

Using the R package “limma,” we merged the AGGLncRNAs with the survival data from PAAD patients. Next, we identified the differentially expressed AGGLncRNAs that were considerably associated with prognosis (*p* < 0.01) by performing the univariate Cox (uni-Cox) regression analysis. Based on the results, we divided the samples randomly into the training and test groups at a 1:1 ratio using the R package “caret”.

Using the R package “glmnet”, we conducted a least absolute shrinkage and selection operator (LASSO) Cox regression analysis and created prognostic signatures by performing multivariate Cox (muti-Cox) regression analysis. The risk score formula was constructed as follows: $RiskScore=\sum_{i=1}^{n}Coef_{i}$ × $X_{i}$, where $Coef_{i}$ denotes the coefficient, and $X_{i}$ denotes the normalized count of each differentially expressed AGGLncRNAs. We separated the PAAD patients into two groups, which included low-risk and high-risk patients. The effectiveness of this signature to predict prognosis was determined using multi-metric ROC curves. We also examined and displayed the ROC curves of the training and test sets for internal validation. The Kaplan–Meier survival curves were used to compare overall survival (OS) between the sets. The prognostic impact of the clinical variables was further validated based on the clinical ROC curves, C-index, and subgroup analysis.

### 2.4. Construction of the Nomogram

To evaluate the risk score as an independent prognostic factor and construct the associated forest plots, we conducted uni-Cox and multi-Cox regression analyses. For the TCGA-PAAD cohort, we used the “rms” package in R to create a nomogram consisting of the risk scores and clinicopathological features for predicting survival after one, three, and five years. To determine whether the nomogram can be used as an independent prognostic predictor, we also performed uni-Cox and multi-Cox regression analyses.

### 2.5. Analysis of Functional Enrichment

Besides conducting the Kyoto Encyclopedia of Genes and Genomes (KEGG) and gene ontology (GO) enrichment analyses, we used the R package “clusterProfler” to thoroughly investigate the functional annotation and pathway analysis of differential genes. We used “c2.cp.kegg.v7.4.symbols.gmt” from MSigDB to perform gene set variation analysis (GSVA). We used the R package “GSVA” to conduct the GSVA enrichment analysis. The R package “heatmap” was used to construct a heat map using the enrichment results. Using the R package “limma”, the differences were considered to be statistically significant at adjusted *p* < 0.05.

### 2.6. The Risk Signature’s Immunity Analysis

We used different algorithms to measure immune infiltration scores. These algorithms included XCELL [24], TIMER [25], QUANTISEQ [26], MCPCOUNT [27], EPIC [28], CIBERSORT [26], and CIBERSORT-ABS [29]. The single-sample GSEA (ssGSEA) method was used to distinguish low-risk patients from high-risk patients based on the immune cell characteristics of the PAAD patients. We also examined the immune checkpoint variations between the patients in the high-risk and low-risk groups.

### 2.7. Analysis of Tumor Mutations

We downloaded the data on the PAAD tumor mutation load (TMB) from the TCGA database. The number of mutated bases per 1 million bases is known as TMB. We collected this data using the Strawberry Perl program. Based on the median TMB value, the data were then sorted into high and low categories. The 15 genes with the highest tumor mutation frequency (TMF) in the PAAD patients from the TCGA database were evaluated and presented using the R package “maftool”. We also analyzed the TMB (*p* < 0.05) in the low-risk group and performed the log-rank test for survival analysis.

### 2.8. Prediction of Targeted Drug Sensitivity and Immunotherapy Response

For patients with PAAD, we applied the tumor immune dysfunction and exclusion (TIDE) score (http://tide.dfci.harvard.edu/ (accessed on 3, October,2022) to predict how they might respond to immunotherapy. Our prognostic model predicted the discrepancy in the chemosensitivity of PAAD patients to common chemotherapeutic agents using the R package “pRRophetic,” which was based on the half-maximal inhibitory concentration (IC50) in PAAD patients obtained from the Genomics of Drug Sensitivity in Cancer (GDSC) database [30].

### 2.9. Statistical Analysis

The R software version 4.2.1 and Strawberry Perl version 5.30.0 were used to conduct the statistical analysis. We compared OS between the high-risk and low-risk groups using Kaplan–Meier (KM) survival curves and log-rank tests. The characteristics were constructed using LASSO-Cox regression models. The predictive ability of the model was assessed using the ROC curve. To evaluate the proportion of tumor-infiltrating immune cells, immune checkpoints, and immune function between the groups, the Wilcox test was conducted; the differences were considered to be statistically significant at *p* < 0.05 and FDR < 0.05.

## 3. Results

### 3.1. Acquisition of Differentially Expressed AGGLncRNAs in PAAD

The flowchart (Figure 1) outlines the main processes of the study. By performing Pearson’s correlation analysis (correlation coefficient >0.4 and *p* < 0.001), we identified 298 AGGLncRNAs (Appendix A). Then, we mapped the associated Sankey plots demonstrating the differential expression for the co-expression relationship with aggrephagy-related genes (Figure 2A).

### 3.2. Construction of a Prognostic Characteristic Risk Score Model for AGGLncRNAs

We created a risk-scoring algorithm based on AGGLncRNAs to more accurately predict the prognosis of PAAD patients. We initially combined the clinical survival data of PAAD patients with the expression of AGGLncRNAs. We obtained 57 AGGLncRNAs associated with the prognosis of PAAD patients by univariate Cox analysis (screening condition: *p* < 0.05) (Appendix A). After searching for lncRNAs with significant associations with PAAD patient outcomes, 57 AGGLncRNAs were used to perform a Lasso regression analysis. Eight lncRNAs were isolated from this and analyzed to determine the regression coefficients and cross-validation trends (Figure 2B–C). Finally, these high-dimensional data were downscaled by a multifactorial Cox proportional risk regression model, and four lncRNAs, including CASC8, LINC01091, LINC02600, and PAN3−AS1, were finally identified. The prognostic index (PI) = (0.607706963762648×ExpressionCASC8) + (−0.867774579447724×ExpressionLINC01091) + (−0.895722611231465×ExpressionPAN3-AS1) + (−0.749903492949133×ExpressionLINC02600). We used the R package “limma” to assess the co-expression relationship between these four OS-associated AGGLncRNAs and ARGs. As shown in the heat map (Figure 2D), UBB, TUBA1C, TUBA1A, PRKN, DYNLL2, and DYNC1LI1 had more correlation compared to the other co-expression relationships. To increase the accuracy and reliability of the analysis, we randomly separated the entire dataset into a training group and a testing group in a 1:1 ratio.

### 3.3. The Survival Analysis of the AGGLncRNAs Signature

To determine whether our established risk score model had a better value for the prognosis of PAAD patients, we evaluated it to determine the survival prognosis. We sorted the PAAD patients into high-risk or low-risk groups. To compare the expected effect of the AGGLncRNAs model, we conducted Kaplan–Meier analysis and plotted survival curves for the entire dataset, test subsets, and training subsets, respectively. The results showed that irrespective of whether the patients were in the all group (Figure 3A), test group (Figure 3B), and training group (Figure 3C), a statistically significant difference occurred between the patients in the high-risk and low-risk groups (*p* < 0.05). The PAAD patients in the low-risk group had longer OS than those in the high-risk group. We constructed heat maps (Figure 3E,F) using the R package “pheatmap”, which showed the distribution of the expression patterns of four lncRNAs associated with OS in the three datasets (all, test, and train) for the patients in the high-risk and low-risk groups. For samples from the low-risk group, three lncRNAs (LINC01091, PAN3-AS1, and LINC02600) were protective, but CASC8 was a risk lncRNA. Thus, LINC01091, PAN3-AS1, and LINC02600 were less expressed in the patients of the high-risk group than in the patients of the low-risk group, where CASC8 was strongly expressed. We ranked the risk scores of PAAD patients in all three datasets and plotted their survival status as a scatter graph. We found that the number of deaths in the high-risk group was high in all three datasets (Figure 3G–L). These results also preliminarily suggested that the model could be used to gauge the prognosis of PAAD patients.

### 3.4. Validation of the Ability of the AGGLncRNA Signature to Predict Prognosis

We used the R packages “survivor” and “survminer” to determine the progression-free survival (PFS) of the total dataset (Appendix A). The outcomes for the patients in the high-risk group, who had considerably fewer years of survival than those in the low-risk group, were found to be as expected. The ROC curves were plotted to measure the efficacy of the model for PAAD patients. The results showed AUC values of 0.705, 0.767, and 0.883 for the one-year, three-year, and five-year survival, respectively, of the PAAD patients (Figure 4A), which indicated the precision and sensitivity of the prediction model for these patients. To avoid bias while evaluating the ROC results for the entire dataset, we also randomly grouped the obtained training and test sets as internal control validation for the ROC analysis (Figure 4B,C). The results suggested that the signature performed well in predicting patient prognosis. To assess the differences among different prediction methods, we constructed ROC curves by combining risk scores and clinicopathological characteristics (Figure 4D). The AUC of the AGGLncRNA signature (0.705) was significantly greater than that of the other clinical parameters. The C-index further demonstrated that the risk score was a superior predictive indication than other clinical decision-making criteria. In conclusion, the AGGLncRNA signature was the optimal choice to predict the prognosis of PAAD compared to other common clinical indicators (Figure 4E).

### 3.5. Survival Analysis of the Clinical Subgroups

To determine whether the prognosis of PAAD individuals differed with the clinical subgroups, we used the R packages “survival” and “survminer” to conduct a clinical investigation of the whole sample subgroup. For the subsequent survival analysis, all samples were separated into subsamples based on age (>65 years and 65 years), gender (male and female), tumor grade (grades I–II and III–IV), pathological N stage (grades N0 and N1), T stage (grades T1–2 and T3–4), and pathological stage (grades I–II and III–IV). Except for the pathological stage III–IV grouping, our results showed that the survival time of high-risk patients was considerably lower than that of low-risk patients in all subgroups (Figure 5). This showed that based on their clinical features, the prognosis of certain PAAD subgroups might also be accurately predicted using the currently identified risk model of the AGGLncRNA signature.

### 3.6. Correlations between Clinicopathological Characteristics and Risk Scores

The survival of patients in the high-risk and low-risk groups differed considerably, determined by analyzing the correlation between clinicopathological variables and prognostic risk scores for the overall cohort. Although the low-risk group had more survivors than the high-risk group, the differences in the clinicopathological traits between the groups were not remarkable (Figure 6 and Appendix A). The risk model outperformed the conventional pathological characteristics concerning the prediction of patient prognosis. The traditional pathological features could not precisely guide the treatment of PAAD patients. To quantitatively assess the condition of the PAAD patients, we constructed a special nomogram.

### 3.7. Construction of a Nomogram Based on Clinical Features

As the constructed risk model was strongly related to a poor prognosis, we merged the OS of PAAD patients and their clinical features in the uni-Cox (Figure 7A) and multi-Cox (Figure 7B) analyses to determine whether our prognostic factors from AGGLncRNAs were independent predictors of survival. The findings showed that the AGGLncRNA signature had independent prognostic predictive power for OS in the entire dataset. We constructed a nomogram (Figure 7C) based on the sex, age, stage, grade, and risk score of the patients to estimate the one-year, three-year, and five-year prognosis survival probabilities of PAAD patients. We used this approach to increase the practical application and usefulness of the established risk model. Using this tool, we analyzed OS quantitatively. Our results showed that the risk score strongly influenced the prediction of OS depending on the nomogram, which further supported the idea that a risk model based on AGGLncRNAs can more precisely indicate the prognosis survival of PAAD patients. Based on the likelihood of one-year, three-year, and five-year OS, the calibration curves showed acceptable agreement for the predicted and observed values (Figure 7D). We performed univariate Cox (Figure 7E) and multivariate Cox (Figure 7F) analyses to determine if the nomogram could be used as an independent predictor of survival. The results showed that it was an independent and accurate predictor of risk.

### 3.8. Assessment of Functional Enrichment

To determine the connection of biological processes and signaling pathways with risk scores, the KEGG enrichment analysis and GO functional analysis were performed to examine differing genes between the classes. Therefore, we performed the KEGG enrichment and GO functional analysis to determine the biological roles linked to risk scores. We used the threshold values of FDR < 0.05 and *p* < 0.05 to select the significantly enriched items. Before performing functional enrichment analysis, we obtained the aggrephagy-related differentially expressed genes (AEGs); the detailed data are shown in Appendix A. In the GO functional ynanalysis, biological processes (BP) mainly included the regulation of hormone levels, trans-saptic signaling control, chemical synaptic transmission modification, and hormone release and delivery. The exterior aspect of the plasma membrane, the T cell receptor complex, the plasma membrane signaling receptor complex, etc., were the key components of the cellular component (CC). Molecular function (MF) includes cation channels, gated channels, and metal ion transmembrane transporters (Figure 8A and Appendix A). KEGG primarily consisted of neuroactive ligand-receptor interconnections, cytokine–cytokine receptor interconnections, the cAMP signaling pathway, cell adhesion molecules, and the T cell receptor signaling pathway (Figure 8B and Appendix A). From the GSVA, 67 highly enriched pathways were discovered (Figure 8C; Appendix A). We found differences in immune-function-related pathways between the low-risk and high-risk categories based on the results of the enrichment function. Then, we conducted a thorough immunological investigation of the two PAAD individual subgroups.

### 3.9. Predicting Immune Cell Infiltration and Response to Immunotherapy Based on the AGGLncRNA Signature

To better visualize whether the AGGLncRNA signature can assess the prognosis of PAAD patients in multiple ways, we performed Spearman’s correlation analysis with various algorithms (TIMER, CIBERSORT, QUANTISEQ, MCPCOUNTER, XCELL, and EPIC) to determine the correlation between risk scores and the number of immune cells in the microenvironment of the PAAD tumor. The final results are shown as a heat map (Figure 9A). The results obtained from each algorithm showed that significantly lower immune cell infiltration occurred in the high-risk group than in the low-risk group, which might be directly related to the poor prognosis of the patients in the high-risk group. We evaluated the ssGSEA scores for immune function because variations in immune cell infiltration frequently cause changes in immune function. Most of the immune function scores were considerably higher in the low-risk category than in the high-risk category (Figure 9B). The results suggested that the risk score signature of AGGLncRNAs can categorize various immune activities and thus, affect how immunotherapy works. Among the results shown in Figure 8B, immunotherapy timing in PAAD patients was predicted by the presence of immune checkpoint molecules. Thus, we investigated the variations in immune checkpoint expression between the cohorts at high and low risk (Figure 9C). Between the high-risk and low-risk subgroups, some immune checkpoint genes were statistically significant. Among them, the expression of TNFSF9 and CD44 was significantly upregulated in the high-risk subgroup, while CD48, LAIR1, BTLA, TNFRSF8, TMIGD2, CD40LG, ADORA2A, HAVCR2, BTNL2, TNFRSF4, CD28, CD200R1, CD200, TNFSF14, IDO2, TIGIT, CD244, CTLA4, LAG3, ICOS, TNFRSF9, CD27, CD86, CD160, and PDCD1 genes were significantly downregulated in the low-risk subgroup. Additionally, these genes served as a foundation for immune checkpoint therapy in PAAD patients. We also used the TIDE algorithm to estimate the likelihood of the effectiveness of immunotherapy (Figure 9D). The TIDE score was higher in the low-risk subgroup than in the high-risk subgroup, which suggested that individuals in the low-risk group had a higher chance of immune escape and a poorer outcome with immunotherapy.

### 3.10. Evaluation of Tumor Mutations

The tumor mutation burden (TMB) is the number of nonsynonymous mutations in the somatic cells within a specific genomic region that can indirectly reflect the ability and extent of neoantigen production by tumors and predict the efficacy of immunotherapy for tumors. Therefore, we further performed an analysis of the tumor mutation burden for the PAAD patients. Mutations were identified in roughly 96.25% of high-risk PAAD patient samples (Figure 10A) and 67.07% of low-risk patient samples (Figure 10B). The top four mutated genes were KRAS, TP53, SMAD4, and CDKN2A in the high-risk and low-risk groups, respectively. The deletion of SMAD4 affected the TGF-β/SMAD4 signaling pathway and made pancreatic cancer more aggressive. Pancreatic cancer patients with SMAD4 mutations showed better therapeutic results when a combination strategy was used, consisting of chemotherapy (S-1) and cintilizumab [31]. TMB was considerably different between the high-risk and low-risk subgroups (Figure 10C), and the mutation rate was significantly lower in the low-risk subgroup than in the high-risk subgroup. These results indicated that the improvement in the prognosis of the low-risk subgroup might be due to the low TMB. To test this hypothesis, we divided the PAAD patients into the high-TMB and low-TMB groups and performed the Kaplan–Meier test (Figure 10D). The survival of the patients in the high-TMB group was significantly lower than that of the patients in the low-TMB group (*p* = 0.008). High TMB might trigger anti-tumor immunity. However, the prognostic survival of the PAAD patients in the low-TMB group was higher. This might be because pancreatic cancer is a type of tumor with low immunogenicity [32,33,34]. Furthermore, the overall survival was higher for the patients in the low-TMB + low-risk group (*p* < 0.001; Figure 10E). Our risk model might predict the prognosis of PAAD patients more accurately when combined with the TMB.

### 3.11. Prediction of Targeted Drug Sensitivity

We conducted a drug sensitivity analysis of 12 common drugs used for treating PAAD patients. The results showed that five drugs (5-fluorouracil, docetaxel, dasatinib, afatinib, and gefitinib) had relatively high IC50 in the low-risk group than that in the high-risk group (Figure 11A,B,E,F,H). We also found that seven drugs (axitinib, cisplatin, erlotinib, irinotecan, oxaliplatin, selumetinib, and gemcitabine) had relatively lower IC50 in the low-risk group than that in the high-risk group (Figure 11C,D,G,I–L). Based on our risk scores, the response of PAAD patients to treatment might be further investigated to improve precise drug therapy.

## 4. Discussion

Although new surgical procedures and pharmaceutical techniques have been implemented in treating PAAD patients, such changes have only slightly improved patient outcomes and prognosis. The discovery of novel biomarkers is excellent for starting the use of precision medicine [4,35]. Morphological variants of pancreatic adenocarcinoma exhibit different histological characteristics. These variations might also have distinct prognoses and different molecular signatures [36,37,38]. The use of traditional clinicopathological features as a guide to patient treatment often results in unsatisfactory outcomes. In the diagnosis of pancreatic adenocarcinoma, the non-specific symptoms related to the tumor and the closeness of the cancerous tissue to major blood vessels result in the late presentation of most cases, either with locally advanced or metastatic disease [39]. Thus, 80–85% of tumors are unresectable at the time of presentation [40]. The long-term survival rates for patients with pancreatic adenocarcinoma are also low. Autophagy affects several critical cellular mechanisms in cancer, whose extensive association with tumors makes it possible to target autophagy for tumor therapy [41]. However, a comprehensive analysis of AGGLncRNAs in PAAD has not been performed. Therefore, we based our analysis on the TCGA-PAAD dataset. In this study, a multi-biomarker prognostic model based on AGGLncRNAs was developed and validated.

We obtained the transcriptomic data, clinical data, and mutation data on PAAD tumors from the TCGA database. We performed the LASSO and COX regression analyses to identify AGGLncRNAs that were strongly associated with the prognosis of PAAD patients. The AGGLncRNAs were used to build a novel prognostic model. To categorize PAAD patients into two distinct prognostic categories, we confirmed that the generated AGGLncRNA signature could be used as an independent predictive factor. The ROC curves and the nomogram were then constructed, and a comprehensive analysis showed that the AGGLncRNA signature had more significant predictive performance compared to other traditional clinical indicators, such as age, gender, grade, and stage. The patients in the low-risk and high-risk groups in the risk model differed significantly in immune correlation analysis, mutation analysis, and drug sensitivity analysis. Our findings provided a theoretical foundation that might help clinicians make decisions and improve the quality of patient survival.

The lncRNA is associated with the growth of malignancy, and it can affect patient prognosis by regulating tumor progression. Thus, it might be used as a new biomarker for tumors [42,43]. In our study, the prognostic model consisted of four AGGLncRNAs, which are closely associated with OS in PAAD patients. These include CASC8, LINCO1091, LINCO_2_600, and PAN3-AS1. The Cancer Susceptibility Candidate 8 (CASC8) is located at 8q24.21 and shows a substantial copy number amplification in cancer tissues [44]. The abnormalities in CASC8 are often closely associated with esophageal squamous cell carcinoma [44], non-small cell lung cancer [45], colon cancer [46], and pancreatic cancer [47]. The RNA-binding protein (hnRNPL) is involved in biological processes, such as the transcription of genes, selective splicing, the processing of protein translations, and chromatin remodeling [48,49,50]. Additionally, hnRNPL is also involved in many oncogenic processes, such as proliferation, anti-apoptosis, and the response and repair of DNA damage [51,52]. CASC8 is an oncogene that can delay the progression of esophageal squamous cell carcinoma by binding directly to hnRNPL and preventing it from being degraded by ubiquitin-mediated processes [44]. Based on the predicted results, we speculated that CASC8 might have a similar role in PAAD to help the development of cancerous tissue and affect the quality of survival of the patients. CASC8 polymorphisms (rs1447295 and rs10505477) are also risk factors for other types of cancer [53,54,55]. Individuals with the rs10505477 TC or CC genotype are less likely to develop cancer than those with the TT genotype [46]. Additionally, rs10505477 disrupts the relationship between CASC8 and the POU5F1BB promoter, which is a cancer susceptibility gene [56,57]. Data on PAN3-AS1, LINCO1091, and LINC02600 are limited, and our study might also contribute to determining these three lncRNAs. This information might provide valuable insights into the treatment of PAAD patients.

The sole member of the type II interferon family is IFN-γ, which signals through its receptors IFNGR1 and IFNGR2 to activate macrophages and promote Th1 responses. It activates multiple innate and adaptive immune responses to induce effective antitumor immunity and eliminate viral infections [58,59]. Natural killer (NK) and natural killer T (NKT) cells can regulate IFN-γ production in innate immunity, whereas CD8+ and CD4+ T cells are the primary paracrine providers of IFN-γ in adaptive immune response [60]. The immune reaction is influenced by IFN-γ, which reprograms macrophages to express the pro-inflammatory M1 phenotype. It helps in “priming” macrophages by boosting their sensitivity to inflammatory chemicals [61]. Some studies have shown that IFN-γ collaborates with Toll-like receptor ligands to stimulate anti-tumor activity in pretreatment macrophages, increase nitric oxide (NO) production, and promote the production of pro-inflammatory molecules, such as TNF and IL-12 family cytokines [62]. By regulating certain gene expression patterns, IFN-γ increases macrophage phagocytosis and the killing of tumor cells (cytokines, chemokine receptors, cell activation indicators, cell adhesion proteins, MHC proteins, etc.) [63]. In the TME, IFN-γ production by cytotoxic immune cells can increase the number of iNOS+ CD206- M1- macrophages, thus decreasing tumor development [64]. The type-II-IFN-response in our study differed considerably between the low-risk and high-risk subgroups (Figure 8B), suggesting that IFN-γ is important for the outcome of PAAD patients. These findings might provide important ideas for conducting detailed studies in the future.

The total number of non-synonymous mutations in somatic cells, including frame-shift mutations, insertions, point mutations, and deletions, is commonly characterized as TMB [65,66]. These mutations are caused by aberrant protein production and might function as novel antigens and stimulate anti-tumor responses [67]. In tumor tissue, several mutations produce many altered peptides, some of which are expressed and processed by the MAJOR histocompatibility complex to produce new antigens against which the immune system can elicit an anti-tumor response. Hence, more DNA mutations are associated with the production of more candidate peptides, which increases the chances of the successful recognition of the neoantigen by the immune system. Several clinical studies have shown that a high rate of clinical benefits in patients with high-TMB tumors treated with immune checkpoint inhibitors (including anti-CTLA-4 treatment for melanoma [68], anti-PD-L1 treatment for uroepithelial cancer [69], and anti-PD-1 treatment for non-small cell lung and colorectal cancer [70,71]) might be associated with a high expression of immunoreactive neoplastic antigens in these tumors [70]. We also conducted a TMB analysis for the survival of PAAD patients and found a significantly lower survival rate in the high-TMB group. Thus, clinicians can selectively administer immune checkpoint inhibitor therapy to PAAD patients based on the TMB grouping described in our study. This might improve the therapeutic outcomes for the patients.

We developed a risk model for predicting and assessing the outcome of PAAD independent of other traditional clinicopathological features. This model might help in the clinical management of patients. This study also provided new insights into the development of new tumor biomarkers. However, our study had certain limitations. First, the study was based on a publicly available database, and more prospective real-world data are needed to validate the clinical utility of our model. Second, as a retrospective study, inherent biases might affect the model. Our conclusions were based only on the analysis of previously collected data, and thus, further experiments are needed to determine the mechanisms by which aggrephagy-related lncRNAs can affect the development of PAAD.

## 5. Conclusions

In this study, we downloaded the expression and clinical data on the RNAs of 178 PAAD patients from the TCGA database. A univariate Cox analysis was performed to identify aggrephagy-related lncRNAs (AGGLncRNAs) associated with PAAD prognosis. The AGGLncRNA signature for PAAD was established by conducting the LASSO analysis. The model was validated by performing Kaplan–Meier analysis and constructing the receiver operating characteristic (ROC) curves. Additionally, we developed a nomogram of risk scores and used it to predict overall survival (OS). Finally, we performed analyses for functional enrichment, relevant immune response, tumor mutation, and targeted drug sensitivity prediction. The risk prognostic model for PAAD patients showed good predictive power and provided new insights into immunopharmacological treatment. Our findings might contribute to the development of accurate immuno-oncology studies on PAAD.

The results of this study were promising for the prognostic prediction of PAAD patients. The expression of lncRNAs associated with aggrephagy in biological specimens of PAAD was not confirmed, and we shall further investigate the expression and the mechanism of action of the four lncRNAs associated with PAAD in the model through wet experiments based on the results of this study. The results of this study need to be confirmed by performing ex vivo experiments. Finally, we need to further investigate the potential mechanistic processes of the prognostic model affecting PAAD to provide a new target for the clinical treatment of PAAD patients and improve the efficacy of systemic therapy. A comprehensive therapeutic approach might enhance the overall survival of PAAD patients.

## Figures and Tables

**Figure 1 genes-14-00124-f001:**
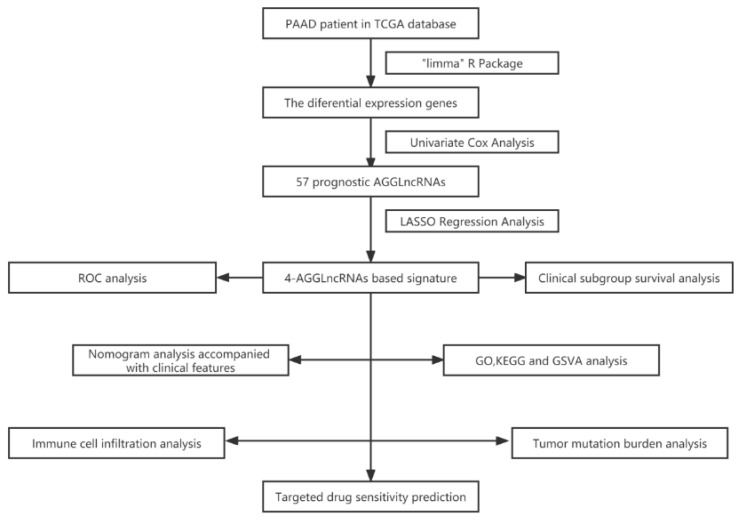
The flowchart shows the design concept of this study.

**Figure 2 genes-14-00124-f002:**
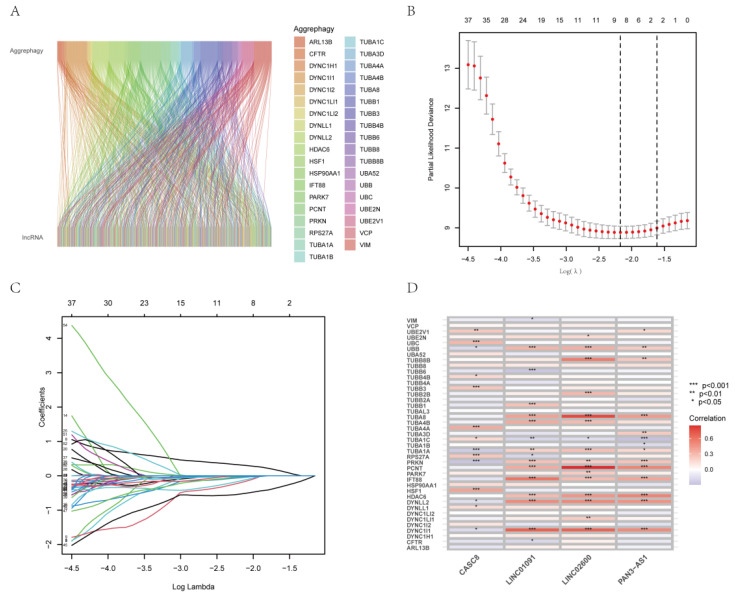
Pancreatic adenocarcinoma aggrephagy-related lncRNA signature construction. (**A**) The Sankey figure shows the correlation between AGGLncRNAs and ARGs. (**B**) In the left diagram, the LASSO Cox regression model shows versus log(λ) of the partial likelihood of deviance. (**C**) The lambda parameter shows the coefficients for certain characteristics. The lambda parameter represents the coefficients of the extracted features. The horizontal coordinate demonstrates the effect on the independent variable lambda and the vertical coordinate denotes the coefficient of the independent variable. (**D**) The heat map depicts the co-expression association between AGGLncRNAs and ARGs; asterisks highlight the degree of significance.

**Figure 3 genes-14-00124-f003:**
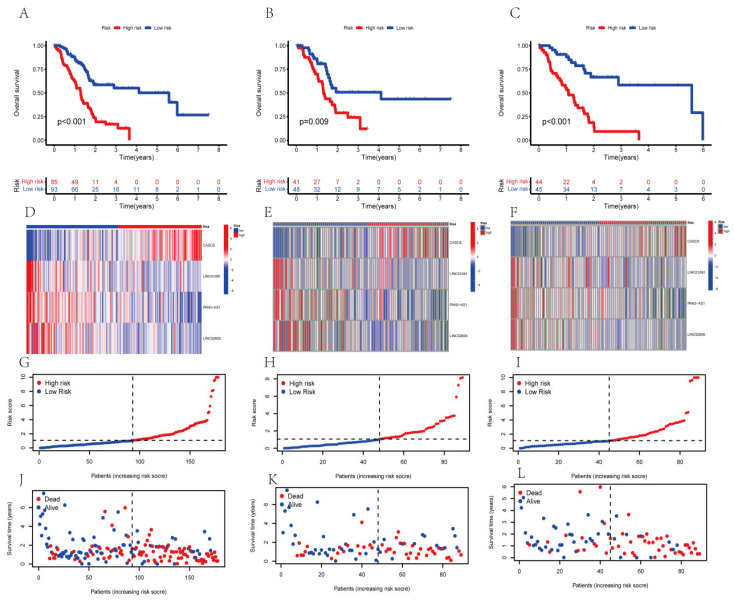
The KM curves of the risk model in the all-data group, training group, and test group. (**A**–**C**) The OS curves of the high-risk and low-risk patients in different groups. (**D**–**F**) The heat map of the expression of the four lncRNAs. (**G**–**I**) The distribution of the risk scores of PAAD patients. (**J**–**L**) The distribution of the survival time and survival status of PAAD patients in the low-risk and high-risk groups.

**Figure 4 genes-14-00124-f004:**
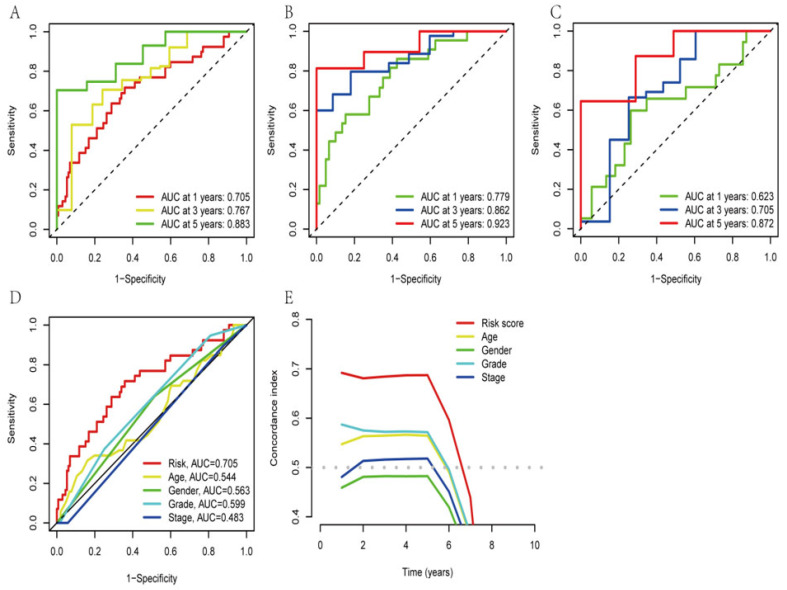
Confirming the signature of the AGGLncRNAs. (**A**–**C**) Time-dependent ROC analysis for one year, three years, and five years, for evaluating the sensitivity and specificity of this prognostic model. A: all, B: train, C: test. (**D**) The ROC curve for the various characteristics. (**E**) The C-index curves of different features.

**Figure 5 genes-14-00124-f005:**
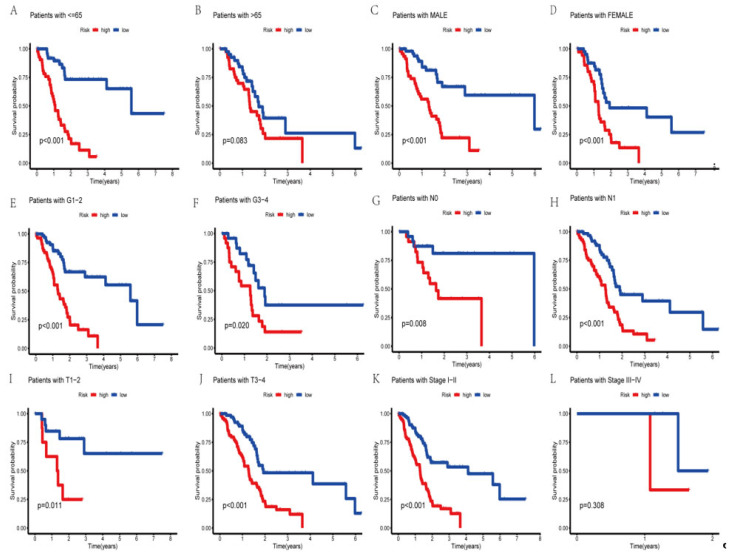
The Kaplan–Meier curves of PAAD patients in the low-risk and high-risk groups, in the context of different characteristics.

**Figure 6 genes-14-00124-f006:**
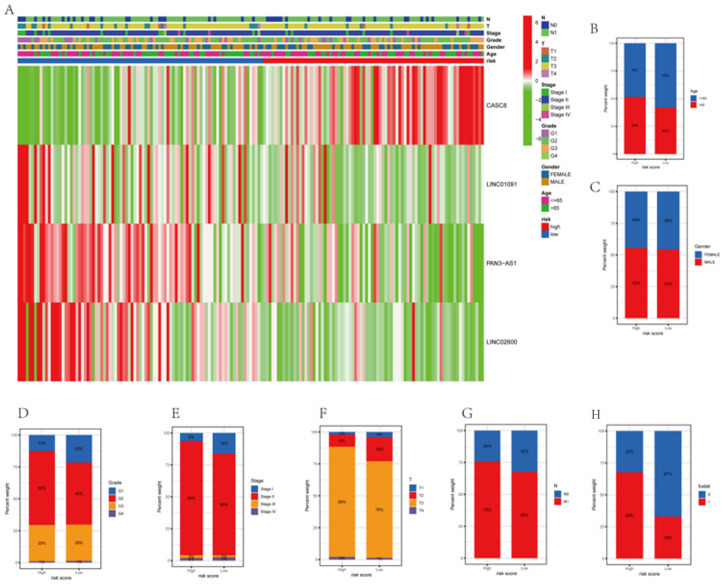
Differences in the clinicopathological characteristics in risk subgroups. (**A**) The heatmap shows the clinicopathological characteristics of patients in high-risk and low-risk groups. (**B**–**H**) The differences between the high-risk and low-risk groups for various pathological traits are presented as histograms.

**Figure 7 genes-14-00124-f007:**
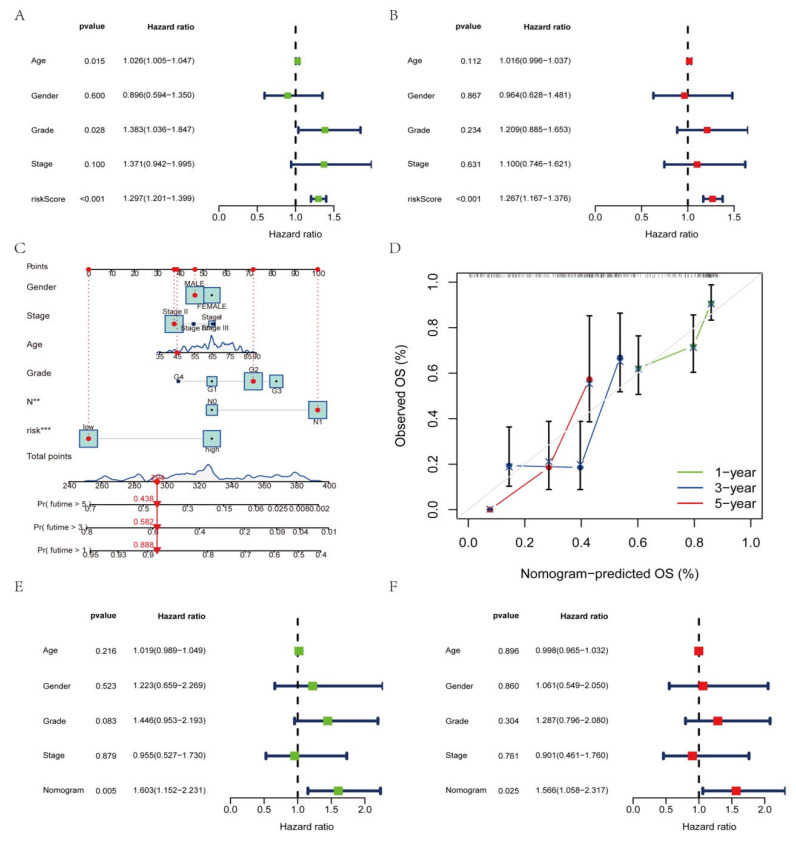
Establishment of the nomogram and evaluation of independent prognostic factors. (**A**,**B**) Uni–COX and multi–COX regression analysis. (**C**) The nomogram for estimating the OS of individuals with PAAD after one year, three years, and five years. (**D**) The calibration curve of the nomogram. (**E**,**F**) Uni–COX and multi–COX regression analysis with the age, gender, grade, stage, and nomogram.

**Figure 8 genes-14-00124-f008:**
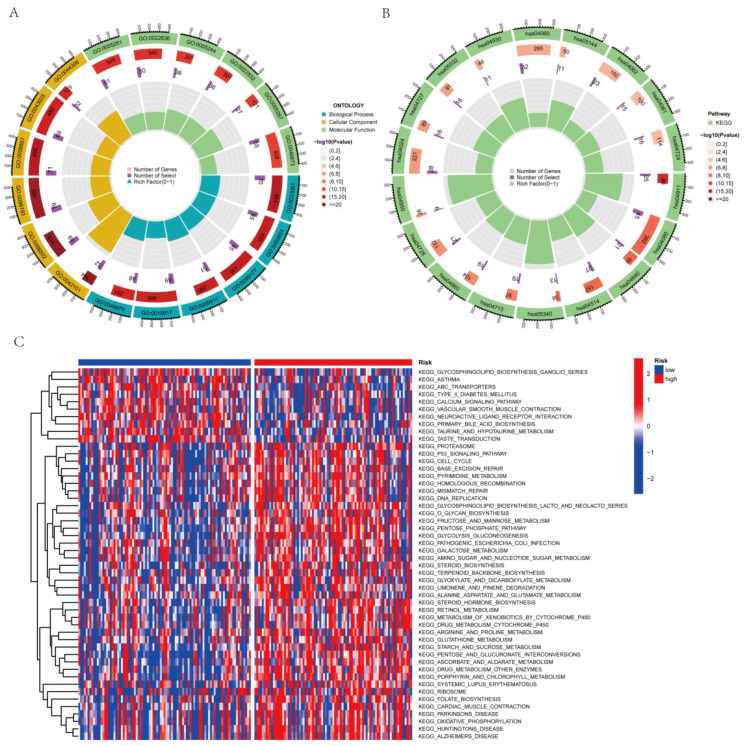
The enrichment pathways in various functional assessments. (**A**) Investigation of the GO enrichment of genes with differential expression. (**B**) The KEGG pathway enrichment investigation of the genes with differential expression (**C**) The GSVA between the high-risk and low-risk cohorts.

**Figure 9 genes-14-00124-f009:**
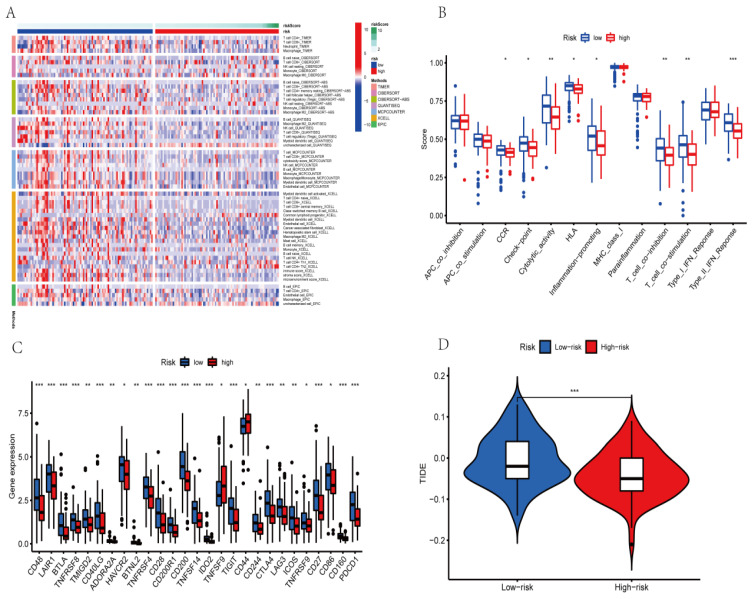
Immune cell infiltration analysis of PAAD patients in high-risk and low-risk groups. (* *p* < 0.05; ** *p* < 0.01; *** *p* < 0.001) (**A**) Immune cell variations between low-risk and high-risk groups based on different algorithms. (**B**) Immune function ssGSEA scores. Differences between low-risk and high-risk groups for (**C**) the immune checkpoints and (**D**) TIDE scores.

**Figure 10 genes-14-00124-f010:**
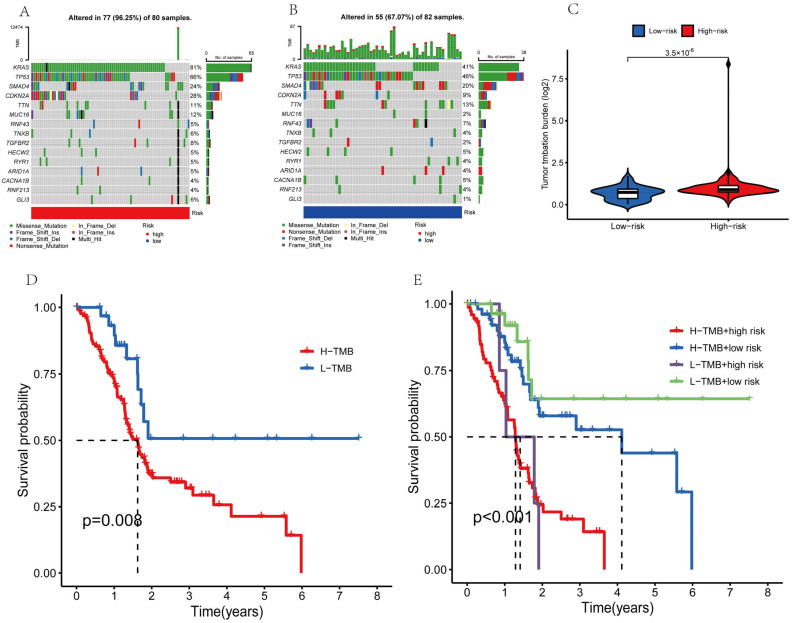
Tumor mutation analysis. (**A**,**B**) The top 15 genes with the most prevalent mutations in the high-risk and low-risk populations. (**C**) Violin plot of TBM. (**D**,**E**) The overall survival curves for different subgroups.

**Figure 11 genes-14-00124-f011:**
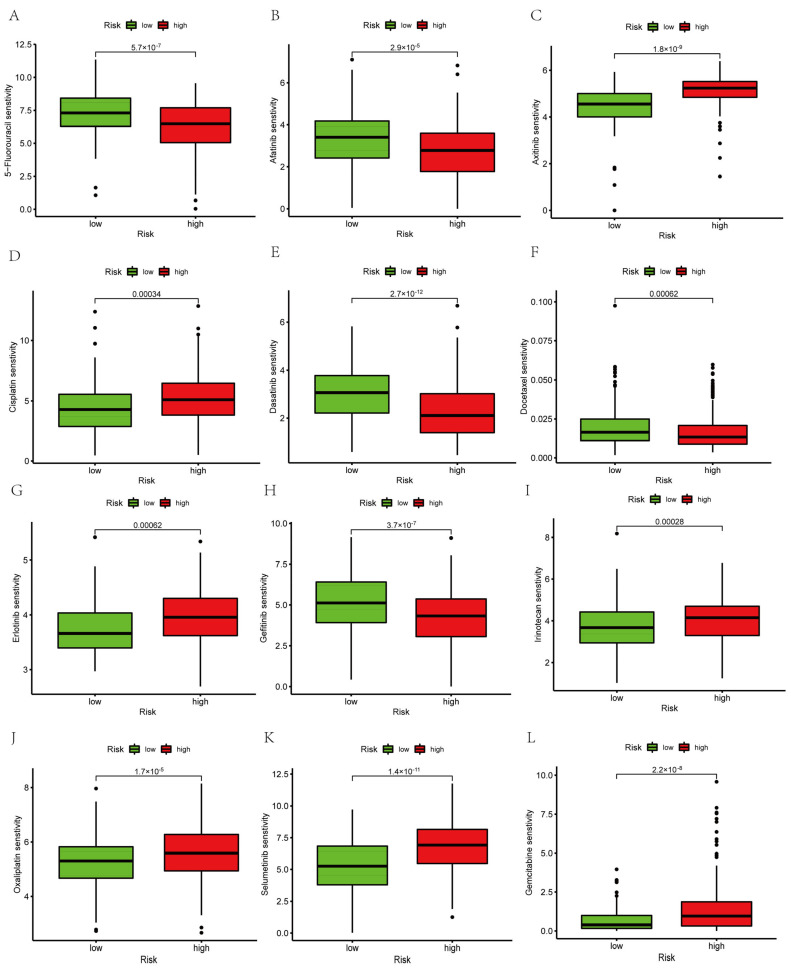
Aggrephagy-Related LncRNA signature predicts targeted drug sensitivity. (**A**) 5−Fluorouracil. (**B**) Afatinib. (**C**) Axitinib. (**D**) Cisplatin. (**E**) Dasatinib. (**F**) Docetaxel. (**G**) Erlotinib. (**H**) Gefitinib. (**I**) Irinotecan. (**J**) Oxaliplatin. (**K**) Selumetinib. (**L**) Gemcitabine.

## Data Availability

The datasets analyzed in the current study are available in the TCGA repository (http://cancergenome.nih.gov/ (accessed on 15 October 2022)). The datasets used and/or analyzed during the current study are available from the corresponding author upon reasonable request.

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
