# Peer review of "An Aggrephagy-Related LncRNA Signature for the Prognosis of Pancreatic Adenocarcinoma"

_genes, 2023, doi:10.3390/genes14010124_

Round 1
Reviewer 1 Report
My opinion:
1. Interesting and current subject matter
2. The graphical presentation of the results increases the value of this publication
Author Response
Dear reviewer:
Thank you for your decision and constructive comments on our manuscript. We have carefully considered the suggestion.We are very grateful for your approval and recognition of our manuscript . Thank you again for your time in reviewing it and we sincerely wish you a Merry Christmas!
Author response: Thank you!
Reviewer 2 Report
Review of:
An Aggrephagy-related LncRNAs Signature for Prognosis of Pancreatic adenocarcinoma
Comments to the Authors
General remarks:
The authors state that the purpose of this study was to investigate the association between the expression patterns of AGGLncRNAs and the prognosis of PAAD patients.
They found that just 4 lncRNAs were enough to forecast the outcome of pancreatic adenocarcinoma. It is not clear to me by what method those were chosen. Please elaborate more on that.
The authors used an extraordinary array of bioinformatic tools during the course of their study.
Specific remarks:
Line 181-185. I don’t understand how the others came up with those 4 lncRNAs as their prognostic markers.
Figure 2B, Table is hard to read.
Figure 2E: Y axis labels are hard to read.
Line2 62-63: This sentence is hard to understand. Please rewrite it more comprehensible.
In general, the resolution of the figures, particularly of the heatmap labels is too low to be readable. Even when zoomed in
Author Response
Dear reviewer:
We thank you for the critical comments and helpful suggestions. We have taken all these comments and suggestions into account,and have made major corrections in this revised manuscript.
We apologize for the poor language of our manuscript. We worked on the manuscript for a long time and the repeated addition and removal of sentences and sections obviously led to poor readability. In addition,we consulted a professional editing service and asked several colleagues who are native English speakers to check the English. We really hope that the flow and language level have been substantially improved.
we sincerely thank the editor and all reviewers for their valuable feedback that we have used to improve the quality of our manuscript. The reviewer comments are laid out below in italicized font and specific concerns have been numbered.our response is given in normal font. In this revised version, changes to our manuscript were all highlighted within the document by using red-colored text.
- “I don’ t understand how the others came up with those 4 lncRNAs as their prognostic markers.”
Response: (Line 183-191)We obtained 57 AGGLncRNAs associated with the prognosis of PAAD patients by univariate Cox analysis (screening condition: p < 0.05) (Figure S1). After searching for lncRNAs with significant associations with PAAD patient outcomes, 57 AGGLncRNAs were used to perform a Lasso regression analysis. Eight lncRNAs were isolated from this and analyzed to determine the regression coefficients and cross-validation trends . Finally, these high-dimensional data were downscaled by a multifactorial Cox proportional risk regression model, and four lncRNAs, including CASC8, LINC01091, LINC02600, and PAN3−AS1 were finally identified.
- “Figure 2B,Table is hard to read.”
Response: (Line 200)This is because the number of LncRNAs is so large that the visualisation is not conducive to reading. We tried several approaches but the final results were not satisfactory. So we have removed this figure from the text. The results have been collated into a three-line table in the supplementary material. This allows the results to be easily viewed.
- “Figure 2E:Y axis labels are hard to read.”
Response: (Line 200) We have rearranged the Y axis labels of this figure to make it better distinguishable. You can find the latest figure in the revised edition.
- “Line2 62-63:This sentence is hard to understand. Please rewrite it more comprehensible.”
Response: (Line 63-65) We are very sorry for the confusion we have caused you through our oversight. We have re-edited the sentence.
- “In general, the resolution of the figures, particularly of the heatmap labels is too low to be readable. Even when zoomed in”
Response: We have reset the resolution of the images in the manuscript to ensure that the image information is easily accessible.
Author response: Thank you!
Reviewer 3 Report
Comments are in the pdf

Author Response
Dear reviewer:
Thank you for your decision and constructive comments on my manuscript. We have carefully considered the suggestion of Reviewer and make some changes. We tried our best to improve and made some changes in the manuscript.
We apologize for the poor language of our manuscript. We worked on the manuscript for a long time and the repeated addition and removal of sentences and sections obviously led to poor readability. In addition,we consulted a professional editing service and asked several colleagues who are native English speakers to check the English. We really hope that the flow and language level have been substantially improved.
we sincerely thank the editor and all reviewers for their valuable feedback that we have used to improve the quality of our manuscript. In this revised version, changes to our manuscript were all highlighted within the document by using red-colored text. Point-by-point responses to the nice reviewer are listed below this letter.
- Line 28
Response:we feel sorry for our carelessness. In our resubmitted manuscript, the typo is revised. Thanks for your correction.
- Line 37
Response : Line 35, Keywords have been in alphabetical order.
- Line 42
Response: Line 41, We have consulted the relevant literature and added the statistics.“Around 331,000 fatalities are caused each year by pancreatic cancer”.
- Line 45
Response: Line 44-45,We have supplemented PAAD patients' 5-year survival rate(only 9% in the USA).
- Line 48-49
Response:Line 47-53,We have reviewed the relevant literature detailing the relationship between the clinicopathological features and the prognosis of patients with PAAD.Early signs of PAAD are not known, and most patients are already at an advanced stage when PAAD is diagnosed. Metastasis to distant organs, such as the liver, lungs, and bones might also occur. The clinicopathological features alone are not effective in guiding the treatment of patients with PAAD to increase their survival. However, since most PAAD patients exhibit different clinicopathological features, this suggests that the traditional pathological grading and staging are not very accurate for guiding the treatment of PAAD patients.
- Line 60-64
Response:Line 62-67,We have reviewed the relevant literature. In other studies, authors used to equate “autophagy” with “Macroautophagy” . Therefore, for ease of scholarly communication, we have also replaced the term “Macroautophagy” with “autophagy” in the manuscript. Aggrephagy is a special type of Macroautophagy (autophagy) . We apologize for the confusion caused by the presentation of this section, so we have rewritten the section.
- Line 68
Response: Line 73, We searched for more literature to support this section. References:“Zhou, J., Li, Y., Liu, X., Long, Y., & Chen, J. (2018). LncRNA-Regulated Autophagy and its Potential Role in Drug-Induced Liver Injury. Ann Hepatol, 17(3), 355-363.” , “Liu, C. Y., Zhang, Y. H., Li, R. B., Zhou, L. Y., An, T., Zhang, R. C., . . . Wang, K. (2018). LncRNA CAIF inhibits autophagy and attenuates myocardial infarction by blocking p53-mediated myocardin transcription. Nat Commun, 9(1), 29.”
- Line 74
Response: Line 77-78, We have added 4 LncRNAs in brackets(CASC8 , LINC01091 , LINC02600 , PAN3−AS1).
- Line 85
Response: In the clinical data of PAAD patients at TCGA, there were 85 patients in stage M0, 5 patients in stage M1 and 86 patients in MX (unknow) stage . The difference in sample volumes between the different M stages was too large to be statistically significant. Therefore we did not include M stage in our study.
- Line 373
Response:Line 393-398,405-408, We have reviewed the relevant literature. For most tumours high-TMB patients predict a better prognosis, which may be related to the immune response and their own anti-tumour immunity. However, the prognostic survival of the low-TMB group in PAAD patients was more promising. This may be related to the fact that pancreatic cancer is a type of tumour with low immunogenicity. We are very grateful for your advice on this. We have revised this section in line with your suggestion. The specific changes have been identified in the text.
- Line 501
Response:Line 526-532, based on your suggestions, we have elaborated on the limitations of our study. First, the study was based on a publicly available database and more prospective real-world data are needed to validate the clinical utility of our model. Second, as a retrospective study, inherent biases might affect the model. Our conclusions were based only on the analysis of previously collected data, and thus, further experiments are needed to determine the mechanisms by which aggrephagy-related lncRNAs can affect the development of PAAD.
- Line 504
Response:Line 547-556, in conclusions, we have added to your suggestions the future perspectives of the study and our plans for further experiments based on the results of the study. The results of this study were promising for the prognostic prediction of PAAD patients. The expression of lncRNAs associated with aggrephagy in biological specimens of PAAD was not confirmed and we shall further investigate the expression and the mechanism of action of the four lncRNAs associated with PAAD in the model through wet experiments based on the results of this study. The results of this study need to be confirmed by performing ex vivo experiments. Finally, we need to further investigate the potential mechanistic processes of the prognostic model affecting PAAD to provide a new target for the clinical treatment of PAAD patients and improve the efficacy of systemic therapy. A comprehensive therapeutic approach might enhance the overall survival of PAAD patients.
Author response: Thank you!